# Molecular Typing of *Listeria monocytogenes* IVb Serogroup Isolated from Food and Food Production Environments in Poland

**DOI:** 10.3390/pathogens10040482

**Published:** 2021-04-15

**Authors:** Beata Lachtara, Jacek Osek, Kinga Wieczorek

**Affiliations:** Department of Hygiene of Food of Animal Origin, National Veterinary Research Institute, 24-100 Pulawy, Poland; beata.lachtara@piwet.pulawy.pl (B.L.); josek@piwet.pulawy.pl (J.O.)

**Keywords:** *Listeria monocytogenes*, serogroup IVb, WGS, cgMLST, virulence genes

## Abstract

*Listeria monocytogenes* is one of the most important foodborne pathogens that may be present in food and in food processing environments. In the present study, 91 *L. monocytogenes* isolates of serogroup IVb from raw meat, ready-to-eat food and food production environments in Poland were characterized by whole genome sequencing (WGS). The strains were also compared, using core genome multi-locus sequence typing (cgMLST) analysis, with 186 genomes of *L. monocytogenes* recovered worldwide from food, environments, and from humans with listeriosis. The *L. monocytogenes* examined belonged to three MLST clonal complexes: CC1 (10; 11.0% isolates), CC2 (70; 76.9%), and CC6 (11; 12.1%). CC1 comprised of two STs (ST1 and ST515) which could be divided into five cgMLST, CC2 covered two STs (ST2 and ST145) with a total of 20 cgMLST types, whereas CC6 consisted of only one ST (ST6) classified as one cgMLST. WGS sequences of the tested strains revealed that they had several pathogenic markers making them potentially hazardous for public health. Molecular comparison of *L. monocytogenes* strains tested in the present study with those isolated from food and human listeriosis showed a relationship between the isolates from Poland, but not from other countries.

## 1. Introduction

*Listeria monocytogenes* is an opportunistic foodborne pathogen responsible for invasive listeriosis, one of the most severe foodborne diseases with a high mortality rate [1]. The infection usually results from the consumption of contaminated food, especially ready-to-eat (RTE) foods of plant and animal origins [2,3]. According to a recent European Food Safety Authority (EFSA) and European Centre for Disease Prevention and Control (ECDC) report, the overall prevalence of *L. monocytogenes* in RTE food of meat origin was 1.4% [1]. Various RTE food categories have different potential to infect consumers with *L. monocytogenes*. For example, deli meats and not re-heated frankfurters were established as food that had a very high predicted risk for consumers due to the relatively high rates of bacterial contamination and ability to support the rapid growth of *L. monocytogenes* under refrigerated storage conditions [4]. Additionally, recent investigations from different countries showed that RTE meat products were a source of epidemics caused by *L. monocytogenes*, including the largest outbreak in South Africa reported so far [5,6].

*L. monocytogenes* has the ability to persist in food processing facilities for months and even years, despite the application of sanitation measures [7,8,9]. Therefore, control of *L. monocytogenes* in the food processing industry is essential to reduce the risk of contamination and to protect consumers [10]. It has also been shown that some *L. monocytogenes* clones survive better in food production environments than others [11]. This phenomenon usually depends on the ability of the bacteria to form a biofilm and express tolerance to sanitizers and other environmental conditions [12]. Furthermore, such strains may also contain genomic islands: Stress Survival Islet SSI-1, which plays a role in acidic and gastric stress responses and growth of the bacteria in food, and SSI-2, involved in oxidative and alkaline stress responses, respectively [13,14,15,16].

Among 13 recognized serotypes of *L. monocytogenes*, only four are of significant public concern, with three, 1/2a, 1/2b, and 4b, being responsible for over 95% of invasive listeriosis cases [17,18]. The majority of the sporadic cases and outbreaks are associated with strains of serotype 4b, while isolates classified to serotypes 1/2a and 1/2c are more often isolated from food and environmental samples [19,20,21]. Based on molecular analyses, *L. monocytogenes* are classified into four lineages, with most isolates belonging to lineages I (serotypes 1/2b, 3b, 3c, and 4b) and II (serotypes 1/2a, 3a, and 1/2c) [17,22]. Furthermore, the serotypes are also distinguished into molecular PCR-based serogroups: IIa (with serotypes 1/2a and 3a), IIb (1/2b and 3b), IIc (1/2c and 3c), IVb (4b, 4d, and 4e) [23].

Classification of *L. monocytogenes* into a certain lineage or serogroup determines some properties of the isolates [24]. For example, the pathogenicity islands LIPI-3 encoding listerolysin S and LIPI-4, containing six genes responsible for a cellobiose-specific phosphotransferase system, are most frequently found among the isolates classified as lineage I. Strains belonging to the IVb and IIb serogroups typically harbour the full length of the *inlA* gene encoding a protein critical for attachment of *L. monocytogenes* to human host cells [17,25]. Additionally, isolates of some serogroups are over-represented and are more often recovered from the same sources, e.g., *L. monocytogenes* of IVb predominates among clinical isolates, including those responsible for meningitis [24,26]. However, classification of *L. monocytogenes* into serogroups is not often enough for epidemiological investigation. In this case, next generation sequencing (NGS) to obtain the whole genome sequence (WGS) is used [6,27,28,29,30,31,32]. WGS-based typing is the preferred method for molecular classification and analyses of *L. monocytogenes* to assess the sources of infection [29]. The broad tools used to analyse the WGS data for the determination of the genetic relationship between isolates are multi-locus sequence typing (MLST) and analysis of core genome (cgMLST), which allow clonal complexes (CCs) with sequence types (STs) and cgMLST complex types (CTs) to be identified, respectively [32,33]. cgMLST for *L. monocytogenes* has been demonstrated as a highly reproducible method and is widely used for molecular typing of isolates classified to the same serotypes and lineages and to identify hypervirulent clones [5,29,31,34,35,36,37].

The objectives of the present study were: (i) to establish the comprehensive molecular characteristics and establishment of the genetic diversity of *L. monocytogenes* IVb serogroup isolated from food and food production environments in Poland; (ii) comparison and assessment of the phylogenetic relationship between the current isolates and strains recovered from food and foodborne listeriosis cases in Poland and in other countries.

## 2. Results

### 2.1. MLST and cgMLST Typing

Among the 91 *L. monocytogenes* isolates of molecular serogroup IVb, three MLST clonal complexes were identified: CC1 (10; 11.0% isolates), CC2 (70; 76.9%), and CC6 (11; 12.1%). CC1 comprised of two STs, i.e., ST1 (9; 9.9% isolates) and ST515 (1; 1.1%), which could be divided into 5 cgMLST. CC2 covered two STs: ST2 (44; 48.3%) and ST145 (26; 28.6%) with a total of 20 cgMLST types, whereas CC6 consisted of only one ST (ST6; 1.1%) classified as one cgMLST (Appendix A).

Based on the cgMLST analysis, the isolates were further classified into three sublineages (SLs: SL1, SL2, and SL6) and 32 CT types, with the most prevalent being CT375 (26; 28.6% isolates) (Table 1). *L. monocytogenes* isolated from RTE food (*n* = 62) belonged to 26 different cgMLST types, mainly to SL2-ST2-CT4325 and SL2-ST2-CT4380 (9 isolates of each; 14.5%). The majority of isolates from raw meat (*n* = 21; 6 cgMLST types) were classified to SL2-ST145-CT375 (16; 76.2%) whereas isolates from food production environments (meat processing plants) (*n* = 8) were diverse (7 different CTs) without any predominant cgMLST type (Table 1).

A clonal relationship of *L. monocytogenes* isolates classified to each of three CCs was determined using a minimum spanning tree (MST) analysis based on cgMLST allelic profiles (Figure 1). It was found that all isolates of CC2, irrespective of origin, were characterized by the lowest number of differences in the allelic variants among the compared sequences, ranging from 0 (no difference = identical isolates) to 48 differences. More heterogenic were *L. monocytogenes* classified to CC1 which differed from 0 to 87 allelic variants, whereas the isolates of CC6 were the most genotypically diverse, ranging from 1 to 110 allelic differences between CTs (Figure 1).

The isolates belonging to the most numerous cgMLST types, i.e., SL2-ST145-CT375 (*n* = 26), SL2-ST2-CT4325 (*n* = 10) and SL2-ST2-CT4380 (*n* = 9) were further compared using MST analysis (Appendix A). Strains classified to CT375 were recovered from all three sources included in the study, mainly from raw meat (n = 16) and RTE food (*n* = 9), during the years 2015–2018 (Appendix A). Interestingly, five of the SL2-ST145-CT375 isolates (IDs: 47103, 47107, 47124, 47125, and 47140) were identical according to the cgMLST allelic profile and were isolated in the years 2016 and 2018 (Figure 1 and Appendix A). More information on the *L. monocytogenes* SL2-ST145-CT375 cgMLST type and other strains shown in Figure 1 and Appendix A are described in Appendix A.

### 2.2. Virulence Factor and Resistance Genes

*L. monocytogenes* isolates were tested towards several virulence marker genes to assess their potential pathogenic hazard to public health (Appendix A). All of them harbored the genes of the pathogenic island LIPI-1, which contains six virulence genes regulated by PrfA regulatory protein, a transcriptional activator for more than 140 genes, including the *inlA* and *inlB* sequences, responsible for *L. monocytogenes* internalization into non-phagocytic cells [38]. Analysis of the *inlA* sequence of all 91 isolates tested revealed that two variants were identified: variant 4 (81; 89.0% strains) and variant 6 (10; 11.0% strains), whereas among the *prfA* gene, four variants were found, i.e., 3 (77; 84.6% strains), 8 (11; 12.1%), 226 (2; 2.2%), and 251 (1; 1.1%), respectively. None of the tested strains possessed the premature stop codons (PMSCs) in the *inlA* gene, involved in attenuation of virulence in *L. monocytogenes* [25,31]. However, the *mdrM* gene, one of the genes of the multidrug resistance transporter MDR playing a role in various drug and bile resistance, was present in all strains tested [32,39]. On the other hand, none of the 91 *L. monocytogenes* isolates possessed the LIPI-4 pathogenicity island and *inlL* internalin gene sequences. WGS data analysis also revealed that some of the isolates harboured only selected virulence genes, e.g., the *inlG* marker was observed in 11 (12.1%) isolates classified to the SL6-ST6 cgMLST profile, whereas 21 (23.1%) strains of ST1, ST6, and ST515 sequence types had the pathogenicity island LIPI-3.

In silico identification of the antimicrobial resistance genes using the Bacterial Isolate Genome Sequence Database *L. monocytogenes* (BIGSdb-*Lm*) revealed that in all 91 *L. monocytogenes* isolates tested four intrinsic genes were present: *fosX* (resistance to fosfomycin), *lmo0919* (lincosamides), *norB* (quinolones), and *sul* (sulfonamides). Furthermore, all strains possessed the phosphatidylglycerol lysyl-transferase (*mprF*) gene, encoding the multiple peptide resistance factor responsible for bacterial peptide resistance [32].

Analysis of the WGS sequences towards the genetic factors responsible for resistance to quaternary ammonium compounds (QACs), including benzalkonium chloride (BAC), revealed that the *Tn6188_qac (ermC)* sequence was present in only one of the strains tested, whereas as many as 68 (74.7%) isolates possessed the Listeria Genomic Island 2 (LGI2) marker. These strains were classified to CC2 (59 out of 62; 95.2% strains) and CC1 (9 out of 10; 90.0% isolates) clonal complexes, respectively. On the other hand, none of the investigated strains harboured the *bcrABC* gene cassette and the *emrE* marker encoding a putative small multidrug-resistant (SMR) efflux pump, both responsible for tolerance to benzalkonium chloride.

Among the two analyzed gene sequences encoding resistance to cadmium (*cadA* and *cadC*), only *cadA* responsible for cadmium-transporting ATPase was identified among 3 of 91 (3.3%) *L. monocytogenes* isolates classified to SL6-ST6 cgMLST type (IDs 47078, 47086, 47110) (Appendix A). However, none of these three isolates had the cadmium and arsenic resistance genes localized on the LGI2 sequence. Further genomic analysis identified another 68 (74.7%) isolates with both cadmium and arsenic resistance sequences present on the LGI2 Island. 

Furthermore, none of the 91 isolates was positive for all five or two genes of Stress Survival Islets 1 or 2 (SSI-1 and SSI-2), respectively, which play a role in bacterial survival under adverse gastric conditions. However, the *lmo0447* gene of SSI-1 was identified in all *L. monocytogenes* strains tested. Additionally, all tested isolates possessed the *comK* gene, responsible for biofilm formation and virulence. Detailed information on all genes identified in the present study are shown in Appendix A.

### 2.3. Detection of Prophage Regions and Plasmid Sequences

Analysis of WGS data of the 91 *L. monocytogenes* isolates revealed a total of 299 DNA prophage sequences, including 69 intact sequences found in 51 (56.0%) strains, with the most common sequence being PHAGE_Lister_vB_LmoS_188, identified among 38 (41.8%) isolates. Furthermore, 132 incomplete and 94 questionable sequences were identified in 87 (95.6%) and 91 (100%) isolates, respectively. Strains with IDs 47078 and 47130 (both from RTE food) and with ID 47084, 47086, and 47110 (originating from food production environments) had the highest number (three in each isolate) of intact prophage sequences (Appendix A).

Examination of *L. monocytogenes* identified 15 (16.5%) strains with plasmid sequences, classified to pLM5578 (12 isolates) and J1776 (3 strains). Isolates which harboured the pLM5578 sequences were classified to CC2, ST2, and four cgMLST types, mainly CT4380 (9 strains). All *L. monocytogenes* with J1776 plasmid belonged to CC6, ST6, and CT434 types and only these strains showed the presence of the *cadA* gene.

### 2.4. Molecular Comparison of L. monocytogenes from Different Sources

cgMLST analysis was used for comparison of the genome sequences of the 91 *L. monocytogenes* tested in the present study with the sequences of 186 *L. monocytogenes* strains available in the BIGSdb-*Lm* database or in GenBank. Detailed information related to these isolates, including the source of isolation and country of origin, are shown in Appendix A. It was found that the current strains classified to clonal complex CC1 did not reveal any genotypic relationship with 59 strains of the corresponding CC, recovered from patients with listeriosis (*n* = 38), food (*n* = 19), and food production environments (*n* = 2) in other countries. Comparison with 20 other *L. monocytogenes* strains previously isolated in Poland, including 14 isolates from clinical cases, showed that some of these, e.g., the current isolates of cgMLST type SL1-ST1-CT322 (IDs 47065, 47070, 47071, 47100, and 47122) and one strain previously described by Kurpas et al. [40] (ID 27929) displayed a very close molecular relationship with up to 7 allelic differences, although they had been isolated in different regions of Poland and in different years. A similar genetic relationship was noted for *L. monocytogenes* ID 47098 and the previously isolated strain ID 27845, both classified to CT4326 and recovered from RTE food, which showed 7 allelic difference in the cgMLST analysis (Figure 2, Appendix A).

Comparative molecular analysis of the present 70 *L. monocytogenes* strains belonging to CC2 and the sequences of 45 other isolates of the same clonal complex did not show any close relationships, especially with the isolates identified in other countries. However, some strains of the current study displayed a genetic similarity with the sequences of other Polish *L. monocytogenes* recovered from food or from clinical cases. For example, four strains of cgMLST type SL1-ST1-CT4382 of RTE food origin revealed a close genetic relationship with the strains of the same origin with IDs 27830, 27800, 27801, 27833, isolated in 2015 and 2016 [40] and with one isolate (ID 34354) responsible for human listeriosis isolated in 2011 [26]. Additionally, 11 strains (IDs: 47080, 47081, 47082, 47083, 47105, 47106, 47111, 47113, 47115, 47119, 47120) from the current study, classified to SL2-ST145-CT375 and mainly recovered from raw meat (10 strains) were identical, based on the cgMLST allelic profile, with strains with ID 27850 and ID 27852 from RTE food previously described in Poland [40]. Similarly, such a close molecular relatedness was also observed among 10 strains classified to SL2-ST2-CT4325, isolated mainly from RTE food and four strains (IDs: 27816, 27834, 27838, 27842) characterized by Kurpas et al. [40] (Figure 3, Appendix A).

Molecular relationships were also identified among *L. monocytogenes* of clonal complex CC6 isolated in Poland but not in other countries, e.g., the strain classified to cgMLST type SL6-ST6-CT434 (ID 47110), isolated in 2016 from food production environments, displayed two to five allelic differences with two isolates (ID 34377, ID 34395) of human origin recovered in 2012 and 2013 [26]. A similar genetic relatedness (from four to six allelic differences) was observed between two current strains (ID 47066 and ID 47067) and four isolates (IDs: 34355, 34383, 34385, 34399) from clinical listeriosis cases from the years 2011–2013 and *L. monocytogenes* SL6-ST6-CT5306 type (ID 47127) of raw meat origin with two human strains recovered in 2011 (ID 34356 and ID 41657) [26] (Figure 4, Appendix A).

## 3. Discussion

The current study on the molecular characteristics of *L. monocytogenes* isolated from food and food production environments are in line with the previous investigations performed in our laboratory [40,41]. However, in contrast to those analyses, the present study focused on the bacteria of serogroup IVb, mainly due to their clinical importance. In Poland, according to the studies of Kuch et al. [26], this serogroup is responsible for more than 55% of invasive listeriosis cases. Additionally, the isolates of food origin usually belong to serogroup IIa or IIb [27,42], thus the information about *L. monocytogenes* IVb may be important to understand the epidemiological chain of food-borne listeriosis. Among the 91 isolates tested in the present study, three cgMLST variants (CC1, CC2, and CC6) were identified, which were also previously found in strains of serogroup IVb [31,32,43]. Isolates of CC6 were described as the cause of meningitis, whereas *L. monocytogenes* classified to CC1 were also isolated from other clinical listeriosis cases [31]. Furthermore, it has been suggested that strains of the later clonal complex show an increased virulence potential as compared to other isolates [31]. However, in the present study, *L. monocytogenes* of CC2 was predominant, which was also found during a previous investigation of food [43]. It seems that this molecular variant may be less virulent than other CCs (e.g., CC4 and CC6) which are mainly responsible for human infections [24,31,43].

Further WGS analysis of the 91 *L. monocytogenes* sequences revealed that the isolates were classified into five sequence types; among them were ST1, ST2, ST6, and ST145, which were previously identified by us in food of animal origin or in food production environments [40,41]. However, the fifth sequence type detected in the current investigation (ST515; one isolate) has not been identified in Poland before. The strains of the four STs mentioned above were recovered from all currently tested sources, which supports the previous findings that most CCs and STs are not assigned to one origin but may be found among various sources [44]. It was described that *L. monocytogenes* isolates of the most common ST2 identified during the present analyses, were responsible for food-borne listeriosis outbreaks worldwide and were also commonly identified in food and food processing environments [32]. Furthermore, *L. monocytogenes* ST6, also commonly detected in the present study, was previously identified as the sequence type involved in food-borne sporadic infections or outbreaks [5,6,30,45].

It has been previously shown that the cgMLST analysis is a very useful molecular tool to assess the *L. monocytogenes* structure population [5,6,27,30,34,35,36]. In the present study, genetically closely related strains of the SL2-ST145-CT375 type were isolated in nine administrative provinces (voivodeships) of Poland during 2015-2018, mostly from raw meat (Appendix A). This fact can be explained by, e.g., the ability of such isolates to persist in food production environments or introduction of the bacteria from outside sources, e.g., from the meat supplying slaughterhouses [35].

Based on the cgMLST results, the current *L. monocytogenes* isolates were compared with the publicly available sequences of Polish and other strains recovered from food, food production environments, and human listeriosis cases. The results showed that national isolates were more closely related to each other as compared to the strains from other countries. This finding supports the data of Lee et al. [43] who suggested that there is a regional molecular heterogeneity among *L. monocytogenes* of the same cgMLST types. On the other hand, some strains isolated previously in Poland from food and listeriosis cases were highly genetically related to the strains of the respective CCs identified in the current investigation [26,40].

The analysis of the WGS sequences of the current isolates towards virulence markers revealed that they were potentially pathogenic for humans since they possessed several genes responsible for, e.g., entering the bacteria into host cells, intracellular replication and escaping from phagocytic vacuoles. The LIPI-1 gene cluster and *inlA* and *inlB* internalin genes were detected in all strains tested, similarly to the results of Camargo et al. [46]. Additionally, within the *inlA* marker, the premature stop codons (PMSCs) responsible for reduced invasion of *L. monocytogenes* were not observed, making the isolate potentially more virulent. The presence of the full-length *inlA* gene among the genome of IVb serogroup isolates was also demonstrated by other authors [25,31,43]. Another internalin family gene member, *inlG*, was identified only in some of the isolates, similarly as in our previous study [40].

In the present investigation, the pathogenicity island LIPI-3 was found in several of the strains classified to clonal complexes CC1 and CC6 but not to CC2. This cluster contains genes involved in the production of listeriolysin S (LLS) and is associated with a higher virulence potential of *L. monocytogenes* due to bactericidal activity and modification of the host microbiota during infection [47]. It was previously shown that such LIPI-3-positive strains, classified to CC6 and ST6, were more often isolated from listeriosis outbreaks [6,30,32,47].

Other molecular markers involved in the pathogenicity of *L. monocytogenes* were identified in the sequences of the currently tested strains. Among them there were, e.g., the arsenic resistance gene cluster of genomic island 2 (LGI2) and the *comK* gene, involved in intracellular survival, biofilm formation and persistence of the bacteria [32,48]. Therefore, identification of such *L. monocytogenes* in food, including RTE food of animal origin, may suggest that they potentially pose a public health risk. It has also been previously shown that strains of IVb serogroup, due to their higher virulence potential, have a reduced ability to survive in food and food production environments compared to other *L. monocytogenes* serogroups [17,24,28]. Indeed, the genetic elements such as *bcrABC* and *Tn6188* (*ermC*) markers associated with BAC tolerance were rare or not present at all among the currently tested strains. Similar results were also obtained by other authors [3,49]. However, the *Tn6188* transposon is often identified in *L. monocytogenes* classified to sequence type ST121, which was not detected in the present study [46,50,51].

In the current investigation, the *cadA* gene, encoding cadmium-transporting ATPase responsible for cadmium resistance, was observed in only few strains classified to SL6-ST6-CT434; all these isolates also harbored the J1776 plasmid sequence. This finding indicates that the *cadA* gene was present on the above plasmid. Other genes encoding resistance to cadmium and arsenic, localized on the LGI2 Island, were identified among 74.7% isolates tested. According to Parsons et al. [52], heavy metals present in the environment can exert a long-term selective pressure on bacteria, including *L. monocytogenes*, and allow them to persist in food or food production environments. Other genetic elements responsible for resistance to various stress conditions were sporadically present in the tested strains. The stress survival islet 1, encoding resistance to a wide range of temperatures, pH or salinity, was represented only by one gene (*lmo0447*) identified in all *L. monocytogenes*, whereas SSI-2, possessing genes responsible for protecting the bacteria against alkaline pH conditions and oxidative stress, was not found in any of the strains tested [13,14,53].

Analysis of the WGS data toward antimicrobial resistance genes revealed that none of the 91 isolates possesses the *penA* (penicillin) and *tetM* and *tetS* (tetracycline) markers. However, the genes responsible for resistance to fosfomycin, quinolones, sulphonamides, and lincosamides were identified. These molecular markers were often detected also in *L. monocytogenes* by other authors [26,34,46,54]. It has been previously described that *L. monocytogenes* and other *Listeria* species are characterized by a low resistance to antimicrobials [36,41]. However, the presence of the antibiotic and sanitizer resistance traits among isolates originated from food and food production environments should be constantly monitored to assess the potential impact on public health.

Over 500 phage sequences present in the *Listeria* genome have been identified, including all *L. monocytogenes* serogroups [55]. In the present study, several intact prophages were detected. Similar results were previously obtained by Matle et al. [54], who found almost the same phages among *L. monocytogenes* isolated from food in South Africa. Such prophage sequences were also reported in strains associated with survival evolution and persistence of *L. monocytogenes* in food-processing facilities [55,56,57]. It has been suggested that prophage sequences present in the *L. monocytogenes* genome were probably one of the main causes of molecular diversity of the isolates classified to the same STs [9,13,58,59]. Furthermore, the presence of some prophages has been connected with the increased virulence and pathogenicity of *L. monocytogenes* [54]. Therefore, identification of prophage sequences in several currently tested strains may suggest the possibility of these isolates acquiring genetic material that would have an influence on a higher infection potential of such *L. monocytogenes* for humans.

## 4. Materials and Methods

### 4.1. L. monocytogenes Isolates

The strains were isolated between 2013 and 2019 during routine microbiological food and food production environment investigations by veterinary official laboratories located in 13 out of 16 voivodeships (administrative provinces) of Poland using the ISO-11290-1 standard method (ISO 11290-1:1996 and 11290-1:2017) and sent to the National Veterinary Research Institute in Pulawy (Appendix A). Then, the isolates were streaked directly on TSYEA (Tryptone Soya Yeast Extract Agar; Bio-Rad, Hercules, CA, USA) and incubated at 37 °C for 24 ± 2 h. The bacteria were stored at −80 °C in a Viabank (BioMaxima, Lublin, Poland). All isolates were then tested toward *L. monocytogenes* molecular serogroups using PCR as described earlier [23,60]. Briefly, *L. monocytogenes* from the Viabank were cultured on TSYEA at 37 °C for 18–24 h and a loopful of bacteria was transferred into 100 µL of TRIS (Tris-(hydroxymethyl)-aminomethane) buffer (A&A Biotechnology, Gdynia, Poland). DNA was isolated using the Genomic Mini protocol (A&A Biotechnology) modified by adding 20 µL of lysozyme (10 mg/mL; Sigma-Aldrich, St. Louis, MO, USA) for 30 min at 37 °C. The amplification reactions were carried out in a thermal cycler (Biometra, Jena, Germany) under the following conditions: initial DNA denaturation at 95 °C for 5 min, followed by 30 cycles of 94 °C for 1 min, 55 °C for 1 min, and 72 °C for 2 min. The final cycle was carried out at 55 °C for 2 min and 72 °C for 5 min.

A total of 1439 *L. monocytogenes* isolates from various sources and voivodeships of Poland were collected. For the purpose of the present study, 91 isolates classified to serogroup IVb and recovered from raw meat (*n* = 21), ready-to-eat (RTE) food of animal origin (*n* = 62), and from food production environments (FPE), i.e., meat processing plants (*n* = 8) were selected and used for further analyzes.

### 4.2. Whole Genome Sequencing (WGS) Analysis

#### 4.2.1. DNA Isolation, Library Preparation and Sequencing 

DNA was extracted as described in point 4.1. DNA quality and concentration were measured by NanoDrop or Qubit 3 (Thermo Fisher Scientific, Waltham, MA, USA). Sequencing libraries were prepared with a Nextera XT DNA Sample Preparation Kit (Illumina, San Diego, CA, USA) and a KAPA HyperPlus Kit (Hoffman-La Roche, Basel, Switzerland) according to the producers’ instructions and sequenced in a MiSeq (Illumina) with a MiSeq Reagent Kit (Illumina) at approximately 50× average coverage. All sequences were trimmed and assembled with Trimmomatic v.0.36 and SPAdes v.3.9.0 [61]. The *L. monocytogenes* sequence parameters used in the present study are shown in Appendix A.

#### 4.2.2. WGS Characteristics of *L. monocytogenes*

MLST (7 loci) and cgMLST profiles (1,748 loci) were extracted from the assemblies using the tool available on the BIGSdb-*Lm* platform [32,33]. MLST profiles with the same alleles for seven loci were classified into sequence types (ST) and grouped into clonal complexes (CCs) if at least five out of seven loci were the same as previously described [33]. cgMLST profiles were grouped into cgMLST types (CTs) and sublineages (SLs), using the cut-offs of seven and 150 allelic mismatches, respectively, as previously described [32]. Allele numbers, CTs, and SLs were determined according to the *Listeria* sequence typing database (BIGSdb-*Lm* platform) [32]. Minimum spanning trees were constructed using BioNumerics software version 7.6 (Applied Maths, Sint-Martens-Latem, Belgium) based on the categorical differences in the allelic cgMLST profiles for each isolate. Loci with no allele calls were not considered in the pairwise comparison between two genomes. The number of allelic differences between isolates was read from genetic distance matrices computed from the absolute number of categorical differences between the genomes.

#### 4.2.3. Identification of Virulence and Other Genetic Markers 

Identification of virulence factor and resistance genes was performed in silico with the Listeria PasteurMLST sequence definition database [32] as described previously by Wieczorek et al. [62]. Detection of particular alleles was based on the virulence, antimicrobial resistance, metal and detergent resistance, stress islands, Listeria Stress Islands, and the *sigB* and rhamnose operon schemes [14,16,32,63,64,65].

#### 4.2.4. Detection of Prophage and Plasmid Sequences

To identify the putative prophage determinants within the genomes of the *L. monocytogenes* tested, the WGS sequences were analysed with the PHASTER (PHAge Search Tool Enhanced Release) web server [66,67]. The presence of plasmid sequences was identified using the PlasmidFinder software 2.1 for the specified Gram-positive scheme [68].

#### 4.2.5. Comparison of *L. monocytogenes* Isolated from Different Sources

A total of 186 *L. monocytogenes* isolates were used for comparison with the current strains. The isolates were selected based on the same CCs as the present strains, i.e., CC1, CC2, and CC6. These *L. monocytogenes* were recovered from humans, food, and food production environments in Poland and in other countries (Appendix A). All sequences meeting the above criteria present in the BIGSdb-*Lm* database (*n* = 168 isolates) were chosen. Additionally, based on the literature [28,69], 18 strains not present in the above data base, but also classified to CC1, CC2, and CC6 and isolated from listeriosis outbreaks, mainly in the U.S. (a total of 10 strains) were selected for the comparison (Appendix A). The *L. monocytogenes* sequences were extracted directly from the BIGSdb-*Lm* database or from GenBank when such *L. monocytogenes* strains were described in the literature [28,40,62,69,70]. Detailed information on these isolates, including the source and year of isolation, country of origin, and accession numbers, are shown in Appendix A. The cgMLST profiles of all compared strains were created using sequences of the 1748 loci according to the scheme described before [32]. The phylogenetic trees, based on cgMLST profiles, were constructed using the BioNumerics 7.6 software as described in point 4.2.2. Altogether, the sequences of 186 *L. monocytogenes* strains were used for phylogenetic comparison, i.e., 79 strains classified to CC1, 46 to CC2, and 62 to CC6, respectively (Appendix A).

#### 4.2.6. Data Availability

All genome sequences of the *L. monocytogenes* isolates used in the present study were deposited in the BIGSdb-*Lm* database under the accession numbers 47065-47116 and 47118-47157.

## 5. Conclusions

WGS data analysis allows the molecular diversity, genetic relationships and identification of pathogenic, survival, and resistance markers of *L. monocytogenes* isolates from food and food production environments to be explored. The present investigation contributes towards a broader information on virulence traits and molecular diversity of *L. monocytogenes* of IVb serogroup isolated from food and food production environments in Poland and in other countries. The cgMLST molecular types with certain virulence gene profiles, especially of the intact *inlA* gene and LIPI-3 pathogenicity island suggest that at least some of the strains are capable of causing human illness. Identification of isolates harbouring the sequences related to stress associated factors and the presence of prophage and plasmid mobile genetic elements enhances the ability of the bacteria to adapt and survive in adverse environmental conditions as well as to increase their pathogenic potential. The presence of genetically identical *L. monocytogenes* recovered from different areas in different years may suggest the ability of such strains to persist outside a host for a long time and/or the cross-contamination of different food production plants. Thus, monitoring and molecular characteristics of *L. monocytogenes* are needed for further improvement of consumers’ safety.

## Figures and Tables

**Figure 1 pathogens-10-00482-f001:**
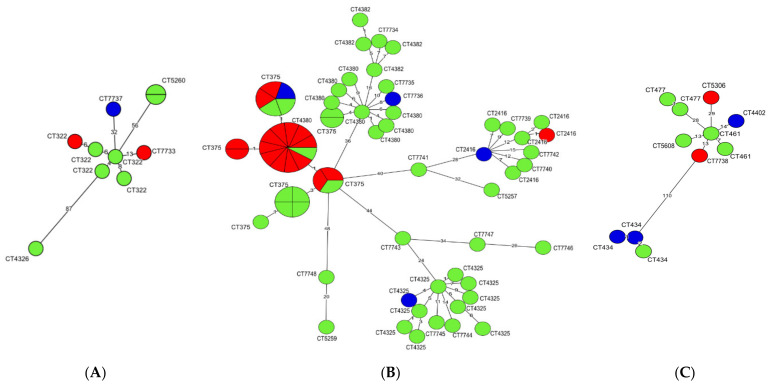
Minimum spanning tree (MST) analysis based on cgMLST allelic profiles of 91 *L. monocytogenes* isolates. Each CC is shown on a separate MST: (**A**) CC1; (**B**) CC2; (**C**) CC6. The circles represent cgMLST types (CTs). The different colors indicate isolate source (green, ready-to-eat food; red, raw meat; blue, food production environments). Numbers on the branches show allele differences between neighboring nodes (CTs).

**Figure 2 pathogens-10-00482-f002:**
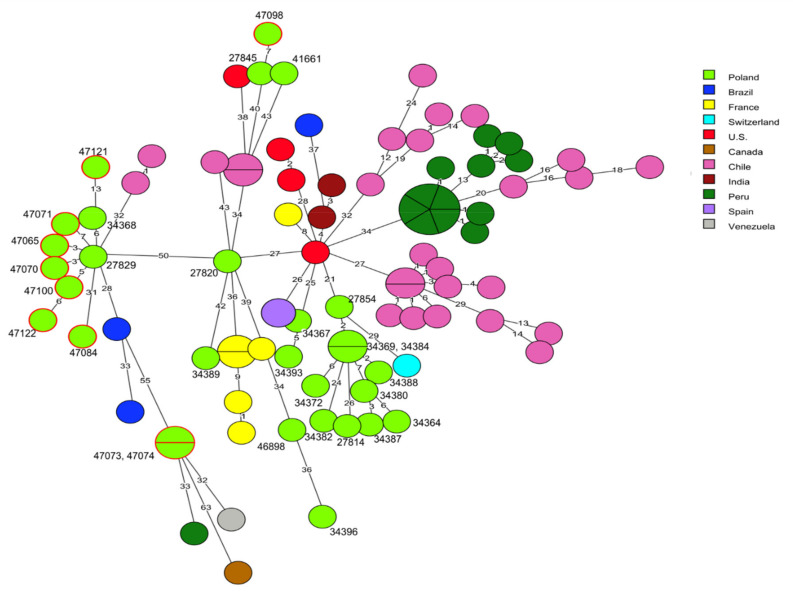
Minimum spanning tree (MST) analysis based on the core genome multi-locus sequence typing (cgMLST) profiles (CTs) of 10 *L. monocytogenes* CC1 strains tested in the present study together with 79 strains of CC1 available at BIGSdb-*Lm* and in the literature. cgMLST types are represented by circles with different colors related to countries of the strains’ origin. Numbers on the connecting lines show alleles differences between adjacent nodes (CTs). The numbers next to circles show ID of Polish *L. monocytogenes* strains. Circles with red rim represent isolates from the present study.

**Figure 3 pathogens-10-00482-f003:**
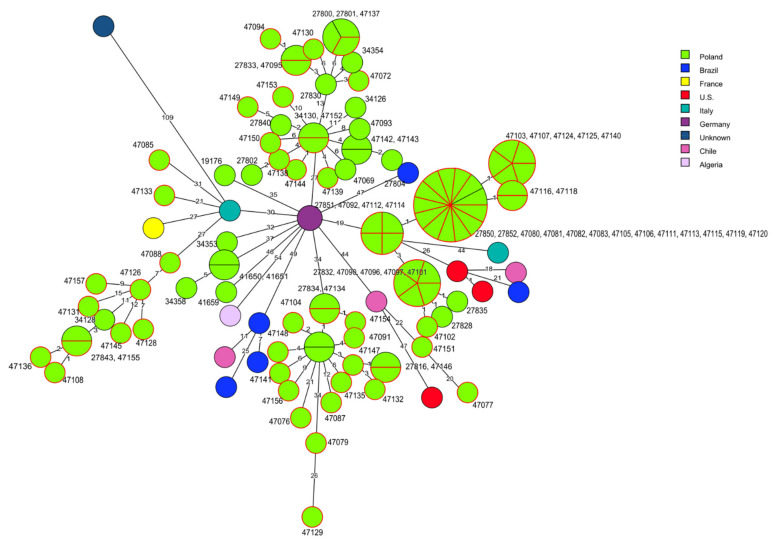
Minimum spanning tree (MST) analysis based on the cgMLST profiles (CTs) of 70 *L. monocytogenes* CC2 strains tested in the present study together with 45 strains of CC2 at BIGSdb-*Lm* and the literature. cgMLST types are represented by circles with different colors related to countries of strains’ origin. Numbers on the connecting lines show allele differences between adjacent nodes (CTs). The numbers next to circles show ID of Polish *L. monocytogenes* strains. Circles with red rim represent isolates from the present study.

**Figure 4 pathogens-10-00482-f004:**
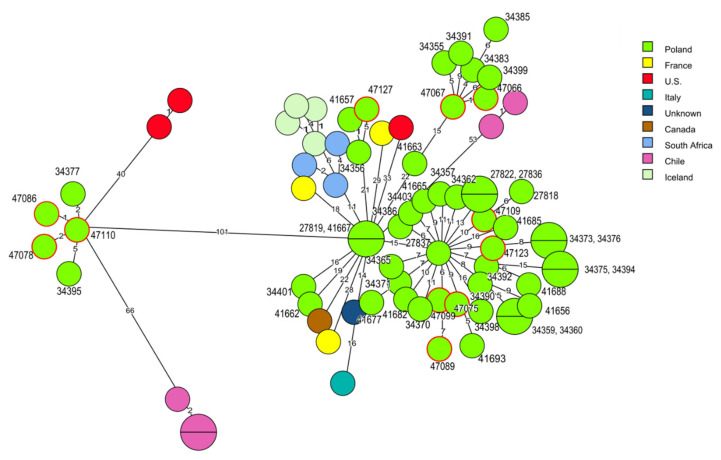
Minimum spanning tree (MST) analysis based on the cgMLST profiles (CTs) of 11 *L. monocytogenes* CC6 strains tested in the present study together with 62 strains of CC6 at BIGSdb-*Lm* and the literature. cgMLST types are represented by circles with different colors related to countries of strains’ origin. Numbers on the connecting lines show allele differences between adjacent nodes (CTs). The numbers next to circles show ID of Polish *L. monocytogenes* strains. Circles with red rim represent isolates from the present study.

**Table 1 pathogens-10-00482-t001:** Molecular characteristics of *L. monocytogenes* serogroup IVb isolates tested.

Strain Origin	Molecular Type (No. of Isolates)			
	Clonal Complex	Sublineage	Sequence Type	cgMLST type
RTE ^1^ (*n* = 62)	CC1 (7)	SL1 (7)	ST1 (7)	CT322 (4), CT5260 (2), CT4326 (1),
	CC2 (49)	SL2 (49)	ST2 (40)	CT4325 (9), CT4380 (9), CT2416 (4), CT4382 (4), CT5257 (1), CT5259 (1), CT7734 (1), CT7735 (1), CT7739 (1), CT7740 (1), CT7741 (1), CT7742 (1), CT7743 (1), CT7744 (1), CT7745 (1), CT7746 (1), CT7747 (1), CT7748 (1)
			ST145 (9)	CT375 (9)
	CC6 (6)	SL6 (6)	ST6 (6)	CT461 (2), CT477 (2), CT434 (1), CT5608 (1)
Raw meat (*n* = 21)	CC1 (2)	SL1 (2)	ST1 (2)	CT322 (1), CT7733 (1)
	CC2 (17)	SL2 (17)	ST2 (1)	CT2416 (1)
			ST145 (16)	CT375 (16)
	CC6 (2)	SL6 (2)	ST6 (2)	CT7738 (1), CT5306 (1)
FPE ^2^ (*n* = 8)	CC1 (1)	SL1 (1)	ST515 (1)	CT7737 (1)
	CC2 (4)	SL2 (4)	ST2 (3)	CT2416 (1), CT4325 (1), CT7736 (1)
			ST145 (1)	CT375 (1)
	CC6 (3)	SL6 (3)	ST6 (3)	CT434 (2), CT4402 (1)

^1^ RTE: ready-to-eat food. ^2^ FPE: food production environments.

## Data Availability

All genome sequences of the *L. monocytogenes* isolates used in the present study were deposited in the BIGSdb-*Lm* database under the accession numbers 47065-47116 and 47118-47157.

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
