# Peer review of "Molecular Typing of Listeria monocytogenes IVb Serogroup Isolated from Food and Food Production Environments in Poland"

_pathogens, 2021, doi:10.3390/pathogens10040482_

Round 1

Reviewer 1 Report

In this paper the authors performed genome analysis of 91 Listeria monocytogenes strains of serotype IVb isolated in Poland from raw meat, RTE food and food production environment between 2013 and 2019.

The genome analysis included determination of CC, ST, cgMLST, virulence and stress-related gene content. Furthermore the authors compared the genome of the 91 strains with other strains, corresponding to the some CC.

The strength of the study is the selection of strains (91 strains of one serotype from a defined source). However, I have several concerns regarding the data presentation and analysis.

Furthermore certain findings are only discussed superficial and the findings are only descriptive. What are the main findings of this study? What is the conclusion?

Abstract:

I would recommend to rewrite the result section of the abstract. The enumeration of the CC, ST, SL and cgMLST is rather confusing. My suggestion is to discuss one CC after the other e.g. CC1 (10;11% isolates) comprised of two STs (ST1 and ST515) which could be dived in 5 cgMLST.

Introduction:

  • line 30-31: Please rephrase, it is misleading
  • line 37: delete significant
  • line43: change infection into contamination
  • Line47: Wrong statement, please revise. SSI-1 has a role in acidic and gastric stress response and growth in food, whereas SSI-2 is involved in oxidative and alkaline stress response.

Results:

  • Please clarify where the strains from FPE come from. Are they from meat processing plants?
  • Table S2: Genome analysis
    • please define the following parameters: min/max/mean/Std
  • MLST and cgMLST:
    • I would recommend to represent the data of one CC after the other (see abstract)
    • Figure 1: The quality of the figure is rather low. Please add the allele differences in the figure rather than using dashed and solid lines.
  • Virulence factors and resistance genes:
    • What was the selection criterium for the virulence and stress-related genes? A list of all analyzed genes and their function would be helpful.
    • I would suggest to but the main findings of the virulence and stress gene analysis in a Figure (see: Maury et al. 2016 or Wagner E 2020 BMC genomics)
    • Table S2: please define the meaning of 1 and 0 in the Table
    • Change the term “positive for” into either “the strains harbor the gene xy” or “gene xy was present”
    • Line133: Is MdrM a stress related or virulence gene?
    • Paragraph Line141-145: a reference is missing
    • Line 158: Have you used the whole SSI-1 (consisting of 5 genes) and SSI-2 (2 genes) for the analysis or only selected genes? If you determined only the presence of lmo0447 (part of SSI-1), than your conclusion is wrong. All strains harbor SSI1. Please change also the discussion accordantly.
  • Prophage region:
    • What conclusion can be drawn from this analysis?
  • Plasmids:
    • Analysis of plasmids is missing!
  • Molecular comparison with other strains:
    • What have been the selection criteria of the included strains? Including other strains for genomic comparison is highly valuable, but clear selection criteria are needed. The number of strains, the location and the source have to be carefully selected to be able to draw any conclusions. My suggestion would be to either compare the strains within one CC of one location (Poland) with different sources (clinical, other food categories, environment) or from different locations, but related to meat.
    • Please highlight the strains used in this study in Figure2-4

Material and Methods:

  • Line 344: Change into “The strains were isolated between 2013 and 2019…
  • Line 354: abbreviation of TRIS is missing
  • Line 364: which type of FPE?

Author Response

Reviewer 1: Thank you very much for the careful reading of the manuscript and for the critical evaluation of our work. All comments were very constructive and we have tried to include all of them during revision of the manuscript. It is maybe possible that, due to a short time received (5 days), not all of your comments were responded as much as you expected. However, we will take into account your valuable suggestions in a future during preparation of next manuscripts on L. monocytogenes. At the moment, we have revised the manuscript as much as possible and all modified sentences have been marked in green in the revised text:

Comment: “Furthermore certain findings are only discussed superficial and the findings are only descriptive. What are the main findings of this study? What is the conclusion?”

Response: More general conclusions have been added into the Conclusion section of the revised manuscript.

 Abstract:

Comment: “I would recommend to rewrite the result section of the abstract. The enumeration of the CC, ST, SL and cgMLST is rather confusing. My suggestion is to discuss one CC after the other e.g. CC1 (10;11% isolates) comprised of two STs (ST1 and ST515) which could be dived in 5 cgMLST”.

Response: Done according to your suggestion.

Introduction:

Comment: Line 30-31: Please rephrase, it is misleading

Response: This sentence has been deleted.

Comment: Line 37: delete significant

Response: The word “significant” has been deleted.

Comment: Line 43: change infection into contamination

Response: The term “infection” has been changed into “contamination”.

Comment: Line 47: “Wrong statement, please revise. SSI-1 has a role in acidic and gastric stress response and growth in food, whereas SSI-2 is involved in oxidative and alkaline stress response”.

Response: The role of Stress Survival Islets has been revised.

Results:

Comment: Please clarify where the strains from FPE come from. Are they from meat processing plants?

Response: The FPE strains were recovered from meat processing plant. This has been clarified in the revised text.

Table S2: Genome analysis

Comment: Please define the following parameters: min/max/mean/Std

Response: We have checked Table S2 and all min/max/mean/Std parameters are shown there. Probably, the problems arise from the layout of the Table S2 on the screen.

MLST and cgMLST:

Comment: I would recommend to represent the data of one CC after the other (see abstract)

Response: Done according to your suggestion.

Comment: Figure 1: The quality of the figure is rather low. Please add the allele differences in the figure rather than using dashed and solid lines.

Response: Fig. 1 has been modified and the dashed lines were changed into solid ones. Furthermore, numbers on the branches were added to show alleles differences between neighboring nodes.

Virulence factors and resistance genes:

Comment: What was the selection criterium for the virulence and stress-related genes? A list of all analyzed genes and their function would be helpful.

Response: In our study, all virulence factor and resistance genes listed in the Listeria PasteurMLST sequence definition database (https://bigsdb.pasteur.fr/cgi-bin/bigsdb/bigsdb.pl?db=pubmlst_listeria_seqdef&page=downloadAlleles) were identified to make the L. monocytogenes analysis comprehensive as much as possible. All these genes and the results of this investigation are shown in Table S1. Detailed information on the virulence and stress-related genes are available under the above link which has been added to the Materials and Methods section.

Comment: I would suggest to but the main findings of the virulence and stress gene analysis in a Figure (see: Maury et al. 2016 or Wagner E 2020 BMC genomics).

Response: WGS analysis of our strains revealed that there are not significant differences in virulence and stress gene profiles. Such differences were noted only for four main gene markers (inlG, cadA, Tn6188_qac ermC, LIPI3). Therefore, we think that a new Figure with the gene profile will not be relevant. Instead, all data related to the presence of virulence and stress genes are described in the text of the manuscript and shown in Table S1.

Comment: Table S2 (should be Table S1): please define the meaning of 1 and 0 in the Table

Response: Explanations for 1 and 0 have been added below the Table S1.

Comment: Change the term “positive for” into either “the strains harbor the gene xy” or “gene xy was present”

Response: The term ,,positive for” has been changed accordingly throughout the whole manuscript.

Comment: Line133: Is MdrM a stress related or virulence gene?

Response: The mdrM gene, one of the multidrug resistance transporter (MDR) genes, confers resistance to various antimicrobial drugs (Tadmor et al., Frontiers in Cellular and Infection Microbiology, 2014, 4, 16) but is also responsible for bile resistance of L. monocytogenes (Moura et al., Nature Microbiology, 2016, 2, 16185). This information has been clarified in the revised manuscript (point 2.2.).

 Comment: Paragraph Line141-145: a reference is missing

Response: The relevant reference [31] has been added.

Comment: Line 158: Have you used the whole SSI-1 (consisting of 5 genes) and SSI-2 (2 genes) for the analysis or only selected genes? If you determined only the presence of lmo0447 (part of SSI-1), than your conclusion is wrong. All strains harbour SSI1. Please change also the discussion accordantly.

Response: The entire SSI-1 and SSI-2 stress survival islets (5 and 2 genes, respectively) were used for the gene presence analysis. None of the 91 isolates was positive for all five or two genes of SSI-1 and SSI-2, respectively. However, the lmo0447 gene of SSI-1 was identified in all L. monocytogenes strains tested. This has been clarified in the revised manuscript (point 2.2.). All results of this analysis are shown in Table S1.

Comment: Prophage region: What conclusion can be drawn from this analysis?

Response: The presence of some prophage sequences have been connected with the increased virulence and pathogenicity of L. monocytogenes, thus, suggesting their higher infection potential for humans. This information has been added at the end of Discussion.

Comment: Plasmids: Analysis of plasmids is missing!

Response: Information about plasmid identification has been added in the Materials and Methods section (point 4.2.4.). Furthermore, in Results, data on the plasmid prevalence has been included in point 2.3. as well as in Discussion. Additionally, in Table S4 detailed information on identified plasmids were added.

Molecular comparison with other strains:

Comment: What have been the selection criteria of the included strains? Including other strains for genomic comparison is highly valuable, but clear selection criteria are needed.

The number of strains, the location and the source have to be carefully selected to be able to draw any conclusions. My suggestion would be to either compare the strains within one CC of one location (Poland) with different sources (clinical, other food categories, environment) or from different locations, but related to meat.

Response: The L. monocytogenes isolates used for comparison with the current strains were selected based on the same CCs as the present strains, i.e. CC1, CC2, and CC6. These isolates were recovered from human listeriosis cases, food, and food production environments in Poland and in other countries. All sequences meet the above criteria, present in the BIGSdb-Lm database (n = 168 isolates) were chosen. Additionally, based on the literature (Burall et al., 2016; Chen et al., 2016), 18 strains not present in the above data base, but also classified to CC1, CC2, and CC6 and isolated from listeriosis outbreaks, mainly in the U.S. (a total of 10 strains) were also selected for comparison with our L. monocytogenes. Such strain selection was done to show all Polish CC1, CC2, and CC6 available sequences in relation to the corresponding strains from other countries and sources. Detailed information related to the selection criteria of the strains used for molecular comparison have been added in the section 4.2.5. of Materials and Methods. The results of these comparisons are shown in Fig. 2 (strains of CC1), Fig. 3 (strains of CC2), and Fig. 4 (strains of CC6). To make the pictures more informed, the isolates of the current study are marked with the red rings in all Figures.
Comment: Please highlight the strains used in this study in Figure 2-4

Response: The L. monocytogenes strains used in the present study have been marked with red circles.

Material and Methods:

Comment: Line 344: Change into “The strains were isolated between 2013 and 2019…

Response: The sentence has been modified accordingly.

Comment: Line 354: abbreviation of TRIS is missing

Response: The abbreviation of TRIS has been expanded and added to the Abbreviation list.

Comment: Line 364: which type of FPE?

Response: The FPE strains were recovered from meat processing plants and this information has been added in the revised manuscript in the Materials and Methods and Results sections.

Reviewer 2 Report

Pag. 4, paragraph 2.4: "cgMLST analysis was used for comparison of the genome sequences of the 91 L. monocytogenes tested in the present study with the sequences of 186 L. monocytogenes strains available in the BIGSdb-Lm database or in GenBank": Specify how these 186 were selected genomes, indicating source of isolation and country of origin, I would suggest adding a table as supplementary material where the access numbers of the strains and related ancillary information are missing in the text.

Pag. 6: "On the other hand, some strains isolated previously in Poland from food and listeriosis cases were highly genetically related to the strains of the respective CCs identified in the current investigation": It is necessary to add an evaluation of this data, for example indicating the country and the origin of the isolates to justify it or to report a supporting bibliographic reference.

Pag. 8: paragraph 4.2.5: specify how these 186 genomes were selected, indicating source of isolation and country of origin.

Pag. 10: Problems displaying the figure were detected, the caption cannot be read and a legend for the colors is missing

Pag. 11: FIGURE 2: The legend, referred to the colors, is outside the margins of the page

Pag. 12: Figure 3: The legend with the colors of the countries of origin is missing

Pag. 13: FIGURE 4: The legend related to the colors is outside the margins of the page

Author Response

Reviewer 2: Thank you very much for the careful reading of the manuscript and for the positive evaluation of our work. According to your comments, we have revised the manuscript and made the modifications marked in red in the revised version of the manuscript.

  1. Introduction has been slightly revised.
  2. Pag. 4, paragraph 2.4: All data about the selected L. monocytogenes genomes used for comparison with the current isolates, including accession numbers, BIGDdb IDs, source of isolation and country of origin, are in Supplementary Table S3 and this information is mentioned in the section 4.2.5. of Materials and Methods.

The Results section has been revised to be more understandable for readers, i.e. Point 2.4, a new sentence has been added “Detailed information related to these isolates, including the source of isolation and country of origin, are shown in Table S3”.

  1. Page 6 (p. 8, L. 277-): "On the other hand, some strains isolated previously in Poland from food and listeriosis cases were highly genetically related to the strains of the respective CCs identified in the current investigation". Response: The relevant references [25,39] have been added (marked in red).
  2. Page 8, paragraph 4.2.5., a new sentence has been added “Detailed information on these isolates, including the source and year of isolation, country of origin, and accession numbers are shown in Table S3”.

Page 10-13: “Problems displaying the figure [Fig. 1] were detected, the caption cannot be read and a legend for the colors is missing”. Further, similar problems were also related to Figs 2-4. We have checked all Figures and all of them show the figures in the correct positions. Probably, the problems arise from the layout of the draft version of the manuscript prepared by the publisher.

Reviewer 3 Report

This is a well conducted and written study.  My only constructive suggestion was for the authors to look over the discussion section to minimize repeating results that were provided in the results section.  From a personal standpoint, I was curious whether there was any correlation between cadmium resistance and sodium arsenite resistance which has also been suggested to be most common in 4b strains.

Author Response

Reviewer 3: Thank you very much for the careful reading of the manuscript and for the positive evaluation of our work.

The whole Discussion has been carefully read again and revised to avoid repeating results that were provided in the results section. The new sequences have been marked in blue in the revised version of the manuscript.

A correlation between cadmium resistance and sodium arsenite resistance, which has also been suggested to be most common in 4b strains, has been analyzed again and this information is now included in the Results and Discussion section of the revised manuscript (marked in blue).

Round 2

Reviewer 1 Report

The authors revised the manuscript according the comments. I have onlay a few minor points:

Results: The quality of the figures should be improved.

Table S2: There is a misunderstanding. I can clearly see the heading in the table. However the following parameters need to be clearly defined. min/max/mean/Std of what? I guess of these parameters correpsond to contigs lengths?

Discussion:

In the current investigation, the cadA gene, encoding cadmium-transporting ATPase 320 responsible for cadmium resistance, was observed in only few strains classified to SL6-321 ST6-CT434; all these isolates also harbored the J1776 plasmid sequence.

This finding indicates that cadA is present on the plasmid…

Author Response

Reviewer 1: Thank you very much for accepting our changes and corrections made after the first revision of the manuscript and for the additional minor comments.

Comment: “The quality of the figures should be improved.

Response: The quality of the figures has been substantially improved.

Table S1

Comment: “There is a misunderstanding. I can clearly see the heading in the table. However the following parameters need to be clearly defined. min/max/mean/Std of what? I guess of these parameters correpsond to contigs lengths?”.

Response: The parameters shown in Table S2 are related to the contigs. The min/max/mean/Std parameters have been clearly defined now. Additionally, the Table S2 name has been slightly changed.

Discussion

Comment: “In the current investigation, the cadA gene, encoding cadmium-transporting ATPase 320 responsible for cadmium resistance, was observed in only few strains classified to SL6-321 ST6-CT434; all these isolates also harbored the J1776 plasmid sequence.”

Response: The suggested sentence: “This finding indicates that the cadA gene was present on the above plasmid” has been added (marked in red in the revised manuscript).
